# The Neuroprotective Flavonoids Sterubin and Fisetin Maintain Mitochondrial Health under Oxytotic/Ferroptotic Stress and Improve Bioenergetic Efficiency in HT22 Neuronal Cells

**DOI:** 10.3390/antiox13040460

**Published:** 2024-04-13

**Authors:** Marie Goujon, Zhibin Liang, David Soriano-Castell, Antonio Currais, Pamela Maher

**Affiliations:** Cellular Neurobiology Laboratory, Salk Institute for Biological Studies, 10010 North Torrey Pines Road, San Diego, CA 92037, USA; zliang@salk.edu (Z.L.); dsorianocastell@salk.edu (D.S.-C.); acurrais@salk.edu (A.C.)

**Keywords:** oxytosis/ferroptosis, mitochondrial dysfunction, bioenergetics, neurodegenerative disease, Alzheimer’s disease, aging, antioxidant defense

## Abstract

The global increase in the aging population has led to a rise in many age-related diseases with continuing unmet therapeutic needs. Research into the molecular mechanisms underlying both aging and neurodegeneration has identified promising therapeutic targets, such as the oxytosis/ferroptosis cell death pathway, in which mitochondrial dysfunction plays a critical role. This study focused on sterubin and fisetin, two flavonoids from the natural pharmacopeia previously identified as strong inhibitors of the oxytosis/ferroptosis pathway. Here, we investigated the effects of the compounds on the mitochondrial physiology in HT22 hippocampal nerve cells under oxytotic/ferroptotic stress. We show that the compounds can restore mitochondrial homeostasis at the level of redox regulation, calcium uptake, biogenesis, fusion/fission dynamics, and modulation of respiration, leading to the enhancement of bioenergetic efficiency. However, mitochondria are not required for the neuroprotective effects of sterubin and fisetin, highlighting their diverse homeostatic impacts. Sterubin and fisetin, thus, provide opportunities to expand drug development strategies for anti-oxytotic/ferroptotic agents and offer new perspectives on the intricate interplay between mitochondrial function, cellular stress, and the pathophysiology of aging and age-related neurodegenerative disorders.

## 1. Introduction

Advances in health practices worldwide are resulting in an increase in the aged population and the emergence of age-related diseases such as dementia across ethnicities and cultures. Yet, neurodegenerative diseases persist as one of the biggest challenges in biomedical science, still lacking sufficient prognostic, diagnostic, and therapeutic tools. Therefore, a deeper understanding of these pathologies is required.

Aging is the major risk factor for developing dementia [1] and old age is associated with dramatic changes in metabolism and energy consumption [2,3]. Age-related mitochondrial dysfunction, including bioenergetic impairments and imbalances in the redox status [4], have been associated with multiple neurodegenerative diseases such as Alzheimer’s disease (AD), Parkinson’s disease (PD), and Huntington’s disease (HD) [5]. Furthermore, mitochondrial dysfunction has now been identified in cases of mild cognitive impairment, as well as dementia [6,7,8], which justifies further research on this organelle as a potential early disease-modifying target.

For the past thirty years, our group has studied oxytosis/ferroptosis, a non-apoptotic regulated cell death pathway characterized by the depletion of the antioxidant glutathione (GSH). The activation of this pathway results in increases in ROS production and lipid peroxidation, as well as calcium influx and, ultimately, cell death [9,10,11]. Not only has this pathway been implicated in age-related neurodegenerative diseases, but it has also led to the development of potent neuroprotective compounds [9,12], some of which are currently in early-stage clinical trials for AD (CMS121 (NCT05318040); J147 (NCT03838185)). Among the neuroprotective compounds discovered using our cell-based phenotypic screening approach are sterubin, from the native California plant Yerba Santa (Eriodictyon californicum) [13], and fisetin, a relatively rare flavonoid found in strawberries [14].

Although the oxytosis/ferroptosis pathway recapitulates many features of mitochondrial dysfunction associated with neuronal cell death [9,11,15,16], the fundamental mechanisms by which mitochondrial signaling affects or drives oxytosis/ferroptosis are still not fully understood. Recently, our group’s work on the non-psychoactive phytocannabinoid, cannabinol (CBN), demonstrated the importance of the preservation of mitochondrial homeostasis in the protection against oxytosis/ferroptosis induced by the glutathione peroxidase 4 (GPX4) inhibitor RSL3 by showing that CBN requires mitochondria to promote cell survival against oxytosis/ferroptosis [17].

Based on these results [17], we hypothesized that the neuroprotection by fisetin and sterubin against oxytosis/ferroptosis might also directly involve mitochondria. To test this idea, we conducted comparative studies of sterubin and fisetin on mitochondrial health in the context of oxytotic/ferroptotic stress in HT22 cells, a hippocampal nerve cell line developed in our lab [18]. In doing so, we aimed at deepening our understanding of mitochondrial physiology in the context of oxytosis/ferroptosis, as well as the pathways driving the neuroprotective effects of these two compounds.

Here, we demonstrate that sterubin and fisetin restore mitochondrial homeostasis in stressed cells. However, in contrast to CBN [17], we found that neither sterubin nor fisetin require mitochondria for their neuroprotective effects, indicating a likely upstream target through which they have a wide range of homeostatic effects. Our data not only further demonstrate the key role of mitochondrial dysfunction in oxytosis/ferroptosis neuronal toxicity but also strengthen the evidence for the therapeutic potential of sterubin and fisetin in the treatment of age-associated neurodegenerative diseases.

## 2. Materials and Methods

Materials: All solvents and reagents were purchased from commercial sources and were used without further purification. Sterubin was obtained from Michael Decker [19], fisetin was obtained from Indofine (Hillsborough Township, NJ, USA), CBN was synthesized in-house, and RSL3 and carbonyl cyanide-4-(trifluoromethoxy)phenylhydrazone (FCCP) were from Sigma-Aldrich (Saint Louis, MO, USA). The Seahorse XFe96 FluxPak (Cat# 102416) and Seahorse XF Cell Mito Stress Test Kit (Cat# 103015) were from Agilent Technologies (Santa Clara, CA, USA). Calcium indicators Rhod-2 AM (Cat# R1244) and Fluo-4 AM (Cat# F14201), CellROX Green Reagent (Cat# C10444), MitoSOX Red Mitochondrial Superoxide Indicator (Cat# M36008), MitoTracker Orange CM-H2TMRos (Cat# M7511), CyQUANT Direct Cell Proliferation Assay Kit (Cat# C35011), Pierce BCA Protein Assay Kit (Cat# 23227), Cell Lysis Buffer (Cat# FNN0011), and SuperSignal West Pico PLUS Chemiluminescent Substrate (Cat# 34578) were from Thermo Fisher Scientific (Waltham, MA, USA). Criterion XT Bis-Tris Protein Gradient Gels (Ca# 3450125), XT MOPS Running Buffer (Cat# 1610788), Trans-Blot Turbo RTA Transfer Kit (Cat# 1704275), and Precision Plus Protein Standards (Cat# 1610373) were from Bio-Rad (Hercules, CA, USA).

General Instrumental Analysis: Optical absorbance and fluorescence were measured on a SpectraMax M5 Multi-Mode microplate reader (Molecular Devices, San Jose, CA, USA).

Microscopy: Brightfield, phase contrast, and fluorescence microscopic images were acquired on an IX51 inverted microscope (Olympus Corporation, Tokyo, Japan) with an INFINITY3 monochrome CCD camera (Teledyne Lumenera, Nepean, ON, Canada) and with ImageXpress Pico Imaging System (Molecular Devices, San Jose, CA, USA). Super-resolution microscopic images were acquired on a Zeiss LSM 880 rear port laser scanning confocal and Airyscan FAST microscope (Carl-Zeiss, Oberkochen, Germany). Image processing and analysis were performed with the microscope software packages ZEN Black and ImageJ/Fiji.(Java 1.8.0_172 (64-bit))

Cell Culture: HT22 mouse hippocampal nerve cells and HT22 mt-GFP and mt-GFP/mCherry-Parkin cell lines, previously generated in our laboratory as described [17,20], were cultured in high-glucose Dulbecco’s modified Eagle’s medium (DMEM) (Invitrogen, Cat# 11995065, Carlsbad, CA, USA) supplemented with 10% fetal bovine serum (FBS) (Invitrogen, Carlsbad, CA, USA) and 1% antibiotics including penicillin and streptomycin (Invitrogen Cat# 10378016, Carlsbad, CA, USA). Cell cultures were incubated at 37 °C in a fully humidified atmosphere containing 10% CO_2_.

### 2.1. Cell Culture Assays

Oxytosis/Ferroptosis: The assay procedure was previously described [13]. HT22 cells were seeded at 3000 cells/well in 96-well tissue culture plates in DMEM plus 10% FBS and 1% antibiotics. After 24 h of plating, the cells were treated with different concentrations of test compounds or a vehicle control alone or with addition of 100 nM RSL3 to induce the cell death cascade. After 16 h of treatment, cell viability was measured by the MTT assay. Optical absorbance was measured at 570 nm on a SpectraMax M5 microplate reader. Samples were analyzed in three to six wells per independent experiment (*n =* 3–6). Results are presented as the percentage of the controls with vehicle alone. Results were verified by visual inspection of the cells under a microscope.

CyQUANT Cell Proliferation: HT22 mt-GFP or mt-GFP/mCherry-Parkin cells were seeded onto 96-well black-walled plates at a density of 4500 cells/well in DMEM supplemented with 10% FBS and 1% antibiotics. After overnight incubation, the cells were pretreated with the mitochondrial uncoupler FCCP at 10 μM or vehicle for 5 h followed by the addition of test compounds or vehicle for an additional 7 h after visual confirmation of mitophagy on a fluorescence microscope (Zeiss LSM 880). After the desired treatments, cells were imaged again prior to adding CyQUANT (Ex/Em = 508/560 nm) reagents and incubation for 1 h. Fluorescence was measured on a SpectraMax M5 microplate reader. Each condition was analyzed in eight wells per independent experiment (*n =* 8). Results are presented as the percentage of the controls with vehicle alone. Results were verified by live-cell imaging under a fluorescence microscope.

ATP assay: The assay procedure was previously described [21]. HT22 cells were plated at 1.5 × 10^5^ cells/dish in 35 mm dishes. After 24 h of culture, the medium was exchanged with fresh medium and the compounds ± RSL3 were added at the concentrations indicated in the figure legend. At the indicated times, the cells were scraped into lysis buffer and assayed for ATP levels using the Invitrogen ATP determination kit (#A22066; ThermoFisher, Waltham, MA, USA) per the manufacturer’s instructions as described previously [21]. ATP levels were normalized to total protein levels in the cell extracts as determined using the bicinchoninic acid (BCA) assay (ThermoFisher).

Reactive Oxygen Species Measurement: The assay procedure was previously described [22]. HT22 cells were seeded onto 96-well black-walled plates at a density of 5000 cells/well in DMEM supplemented with 10% FBS and 1% antibiotics. After the desired treatments, mitochondrial superoxide was detected with MitoSOX Red reagent (Ex/Em = 510/580 nm). Fluorescence was measured on a SpectraMax M5 microplate reader. Data were normalized for total protein/well. Each condition was analyzed in eight to sixteen wells per independent experiment (*n =* 8–16). Results are presented as the percentage of the controls with vehicle alone. Results were verified by live-cell imaging under a fluorescence microscope.

Calcium Measurements: The assay was performed as previously described [23]. Briefly, HT22 cells were seeded onto 96-well black-walled plates at a density of 5000 cells/well in DMEM supplemented with 10% FBS and 1% antibiotics. After the desired treatments, Ca^2+^ levels were detected with Fluo-4 AM (Ex/Em = 494/516 nm) and Rhod-2 AM (Ex/Em = 552/581 nm) calcium indicator dyes specific to cytosol and mitochondria, respectively. Fluorescence was measured on a SpectraMax M5 microplate reader. Data were normalized for total protein/well. Each condition was analyzed in four to twelve wells per independent experiment (*n =* 4–12). Results are presented as the percentage of the controls with vehicle alone. Results were verified by live-cell imaging under a fluorescence microscope.

Mitochondrial Mass Measurement: The assay procedure was previously described [24]. HT22 cells were seeded onto 96-well black-walled plates at a density of 5000 cells/well in DMEM supplemented with 10% FBS and 1% antibiotics. After the desired treatments, MitoTracker Orange CM-H2TMRos (Ex/Em = 554/576 nm) and Hoechst 33342 (Ex/Em = 360/460 nm) dyes were added to the cells and incubated under the same culture condition for 2 h. Fluorescence was measured on a SpectraMax M5 microplate reader. Data were normalized for total protein/well. Each condition was analyzed in ten wells per independent experiment (*n* = 10). Results are presented as the percentage of the controls with vehicle alone. Results were verified by live-cell imaging under a fluorescence microscope.

### 2.2. Bioenergetics Analysis

Seahorse XF Analysis: The assay procedure was previously described [22]. Cellular oxygen consumption rates (OCRs) were assayed with an XF Cell Mito Stress Test Kit using a Seahorse XFe96 Extracellular Flux Analyzer (Seahorse Bioscience, North Billerica, MA, USA). Complete Seahorse XF DMEM assay medium was supplemented with 10 mM glucose, 1 mM pyruvate, and 2 mM L-glutamine, at pH 7.4. Mitochondrial electron transport chain (ETC) inhibitors were used for sequential injections at the following concentrations: 1.5 μM oligomycin, 2 μM FCCP, and 0.5 μM of a 1:1 mixture of rotenone and antimycin A. Total ATP production and ATP production rate from both glycolytic and mitochondrial pathways were assayed using an XF real-time ATP rate Test Kit using a Seahorse XFe96 Extracellular Flux Analyzer (Seahorse Bioscience, North Billerica, MA, USA). Complete Seahorse XF DMEM assay medium was supplemented with 10 mM glucose, 1 mM pyruvate, and 2 mM L-glutamine, at pH 7.4. In this assay, 1.5 μM oligomycin and 0.5 μM of a 1:1 mixture of rotenone and antimycin A were used for sequential injection. Measurement of the contribution of individual parameters for non-glycolytic acidification, glycolysis, glycolytic capacity, and glycolytic reserve was achieved with the XF Glycolysis Stress Test Kit using a Seahorse XFe96 Extracellular Flux Analyzer (Seahorse Bioscience, North Billerica, MA, USA). For the XF Glycolysis Stress Test, complete Seahorse XF DMEM assay medium was supplemented with 2 mM L-glutamine. Following basal extracellular acidification rate (ECAR) measurement, cells were sequentially injected with 10 mM glucose, 1 μM oligomycin, and 50 mM 2-deoxy-D-glucose (2-DG). All three analyses were conducted using Wave software and XF Report Generators (Agilent Technologies, Version 2.6.3.5). The sensor cartridge for the XFe analyzer was hydrated overnight at 37 °C before the experiment. OCR data were normalized for total protein/well. Each condition was analyzed in sixteen wells per independent experiment (*n =* 16). For HT22 cells, 3000 cells/well were seeded onto the Seahorse XFe96 plates under standard culture condition as described above. The next day, cells were pretreated with fisetin or sterubin at the desired concentrations for 1 h followed by addition of RSL3 (50 nM) and coincubation for 16 h. Immediately before the assay, the culture medium in the plates was replaced with complete Seahorse XF DMEM assay medium. The plates were incubated for 1 h at 37 °C prior to each of the three XF tests performed according to the manufacturer’s instructions.

### 2.3. Morphological Analysis

Mitochondrial Network Morphology Analysis: HT22 mt-GFP cells were seeded at 10,000 cells/well on glass coverslips in 24-well plates in DMEM supplemented with 10% FBS and 1% antibiotics. The following day, the cells were treated with the test compounds for the desired period of time. After rinsing with PBS, the cells on glass coverslips were fixed with 4% paraformaldehyde (pH 7.4) for 10 min at 37 °C. After additional PBS rinses, the coverslips were mounted onto glass microslides with Fluoro-Gel (Electron Microscopy Sciences, Hatfield, PA, USA, Cat# 17985–10). Z-stack images of fixed cells were acquired on a Zeiss LSM 880 rear port laser scanning confocal and Airyscan FAST microscope with ZEN Black software (system 2.3) to trace and render cells in 3D. The mitochondrial network morphology parameters (*n* = 12 cells/condition) were scored and analyzed with the MiNA module of ImageJ/Fiji software (Java 1.8.0_172 (64-bit)) using previously described methods [25].

### 2.4. Western Blots

Sample Preparation: The procedure was carried out as described [26]. HT22 cells were seeded at 3 × 10^5^ cells/dish in 60 mm dishes in DMEM supplemented with 10% FBS and 1% antibiotics, and then exposed to the desired treatments. For total protein extracts, HT22 cells were washed with ice-cold phosphate-buffered saline (PBS) and lysed with cell extraction buffer containing 10 mM Tris, pH 7.4, 100 mM NaCl, 1 mM EDTA, 1 mM EGTA, 1 mM NaF, 20 mM Na_4_P_2_O_7_, 2 mM Na_3_VO_4_, 1% Triton X-100, 10% glycerol, 0.1% sodium dodecyl sulfate (SDS), 0.5% sodium deoxycholate, a protease inhibitor cocktail, and a phosphatase inhibitor cocktail followed by centrifugation (14,000× *g*) for 30 min at 4 °C. The resulting supernatants of total protein extracts were stored at −80 °C until analysis. Concentrations of the harvested proteins were determined by the BCA protein assay.

Western Blotting: Immunoblotting was carried out as described [13]. Laemmli sample buffer with 2% β-mercaptoethanol was added to the samples, and then boiled for 5 min prior to SDS-PAGE. Equal amounts of cellular protein for each sample (10 μg per lane) were resolved by 4–12% gradient Criterion XT Precast Bis-Tris Gels (Bio-Rad, Hercules, CA, USA) and transferred onto PVDF membranes with a Trans-Blot Turbo System (Bio-Rad, Hercules, CA, USA). The membranes were blocked with 5% skim milk in TBST (20 mM Tris buffer, pH 7.5, 0.5 M NaCl, and 0.1% Tween 20) for 1 h at room temperature and incubated overnight at 4 °C with the diluted primary antibody in 5% BSA in TBST. After washes with TBST, the membranes were incubated with horseradish peroxidase-conjugated goat anti-rabbit or anti-mouse secondary antibodies (Bio-Rad, Hercules, CA, USA) diluted 1:5000 in 5% skim milk in TBST. After additional washing, protein bands were detected using the SuperSignal West Pico PLUS Chemiluminescent Substrate, and visualized with a ChemiDoc MP Imaging System (Bio-Rad, Hercules, CA, USA). Band density was quantified densitometrically and normalized to Stain-Free total protein with Image Lab software (Bio-Rad, Hercules, CA, USA). Normalization was performed by measuring total protein directly on the membrane that was used for Western blotting. Using the Image Lab software, the total density for each lane was measured from the blot and a lane profile was obtained. The background was adjusted in such a way that the total background was subtracted from the sum of density of all the bands in each lane. Each Western blot was performed three to six times with independent protein samples (*n =* 3–6). The primary antibodies used were from the following suppliers: antibodies against OPA1 (Cat# 80471, 1:3000), MFN2 (Cat# 9482, 1:3000), DRP1 (Cat# 8570, 1:3000), MFF (Cat# 84580, 1:3000), TOM20 (Cat# 42406, 1:3000), SIRT1 (Cat# 9475, 1:3000), total AMPKα (Cat# 5831, 1:3000), phosphoThr172- AMPKα (Cat# 2535, 1:1000), MCU (Cat# 14997, 1:3000), and NQO1 (Cat# 62262, 1:3000) were from Cell Signaling Technology (Danvers, MA, USA); antibodies against HO–1 (Cat# SPA-896, 1:3000) was from Stressgen (Victoria, BC, Canada); and anti-PGC-1α (Cat# AB3242, 1:3000) was from Sigma-Aldrich (Saint Louis, MO, USA). Horseradish peroxidase-conjugated secondary antibodies (Cat# 1706515, and 1706516) were from Bio-Rad (Hercules, CA, USA).

### 2.5. Statistical Analysis

Data from a minimum of three independent experiments were normalized, pooled, and analyzed using Excel and Graph Pad Prism 9, followed by the statistical tests indicated in the figure legends. Data are presented as the mean ± SD. The data were analyzed by one-way ANOVA with Tukey’s multiple comparison post hoc test or Student’s *t*-test where appropriate. *p* values less than 0.05 were considered statistically significant (* *p* < 0.05, ** *p* < 0.01, *** *p* < 0.001, and **** *p* < 0.0001).

## 3. Results

### 3.1. Sterubin and Fisetin Are Effective Inhibitors of RSL3-Induced Oxytotic/Ferroptotic Cell Death and Specifically Suppress Mitochondrial ROS Increase during Oxytosis/Ferroptosis

Sterubin [13] and fisetin [9] were both previously found to provide significant protection against oxytosis/ferroptosis induced by RSL3, a glutathione peroxidase 4 (GPX4) inhibitor [16], in mouse HT22 hippocampal neuronal cells. As shown in Figure 1B,C, sterubin and fisetin alone did not induce visual changes in cellular morphology as compared to the control HT22 cells. However, treatment with RSL3 for 16 h led to dramatic changes in cellular morphology with large numbers of rounded, shrunken, and detached cells as compared to the control group (Figure 1D). Such phenotypic changes are characteristic of the induction of oxytosis/ferroptosis and are consistent with previous findings [17]. The co-treatment of RSL3 with sterubin or fisetin effectively prevented these morphological changes, as the majority of HT22 cells appeared healthy with a good adherence and fine projections similar to what is seen in the control conditions (Figure 1E,F). In line with this, dose–response measurements of cell viability showed that fisetin and sterubin at concentrations around the previously estimated half-maximal effective concentration (EC50) values [13,27] robustly inhibited the cell death induced by 100 nM RSL3 (Figure 1G,H). Based on these results, the following studies used concentrations of 2.5 μM for sterubin and 5 μM for fisetin.

A key aspect of the oxytosis/ferroptosis pathway is the induction of intracellular oxidative stress by ROS released by mitochondria [9,28,29]. Mitochondrial ROS (mtROS) subsequently contributes to lethal lipid peroxidation [30]. We have previously shown that both fisetin and sterubin can prevent increases in oxidative stress in response to treatment with the oxytosis/ferroptosis inducer glutamate [13,31]. Here, we asked if fisetin and sterubin could also reduce the increase in mitochondrial ROS production seen in response to RSL3. We used MitoSOX, a ROS superoxide fluorogenic probe specifically targeting mitochondria in live cells, for these studies. As shown in Figure 2A, HT22 cells treated with RSL3 for 16 h showed a 58% increase in the MitoSOX signal in comparison to the control cells (*p* < 0.0001). Importantly, the co-treatment with sterubin or fisetin strongly and significantly suppressed the RSL3-induced ROS production from mitochondria (*p* < 0.0001). We also observed a much smaller but significant increase in the MitoSOX signal following a 16 h treatment with fisetin or sterubin alone (*p* < 0.0001).

To extend our investigation on the antioxidant effects of fisetin and sterubin, we measured the levels of two proteins known to be involved in endogenous antioxidant defenses by Western blotting, heme oxygenase-1 (HO-1) and NAD(P)H Quinone dehydrogenase 1 (NQO1) [32]. Both of these proteins are downstream of the antioxidant transcription factor NRF2, which we previously found to be upregulated by sterubin [13] and fisetin [31]. Immunoblotting (Figure 2B,C) showed that the compounds alone were able to strongly stimulate the upregulation of HO-1 and NQO1 relative to control cells, as well as in cells treated with RSL3.

Together, these results demonstrate that the compounds can specifically lessen the ROS production from mitochondria and induce antioxidant defenses under conditions of oxytotic/ferroptotic stress.

### 3.2. Sterubin and Fisetin Promote Mitochondrial Calcium Homeostasis during Oxytosis/Ferroptosis

Maintaining mitochondrial calcium (Ca^2+^) homeostasis is crucial for various neuronal functions. Excessive Ca^2+^ influx in mitochondria increases mtROS production and has been linked to oxytosis/ferroptosis [23,33]. We, therefore, investigated how sterubin and fisetin impact mitochondrial Ca^2+^ homeostasis during oxytotic/ferroptotic stress. As shown in Figure 3A,B, treating HT22 cells with fisetin and sterubin alone overnight for 16 h did not alter Ca^2+^ influx into either the cytosol or mitochondria, as indicated by Fluo-4 AM and Rhod-2 AM calcium indicators, respectively. Conversely, overnight exposure to RSL3 led to a significant rise in Ca^2+^ levels in both cellular compartments (*p* < 0.05), which was counteracted by fisetin and sterubin treatment (*p* < 0.0001).

The mitochondrial calcium uniporter (MCU) serves as a pivotal calcium channel situated in the mitochondrial inner membrane, facilitating the intake of Ca^2+^ ions into the mitochondrial matrix. The dysregulation and malfunction of the MCU have been associated with mitochondrial bioenergetic impairment and, to a larger extent, with neurodegenerative disorders [34]. Therefore, we also investigated the effects of our compounds on MCU protein levels. A 16 h treatment with fisetin or sterubin alone did not affect the basal MCU expression in the HT22 cells (Figure 3C,D). However, while RSL3 treatment significantly upregulated MCU in the cells (231% of control, *p* < 0.001), both compounds prevented this increase. Sterubin (95% of control) appeared somewhat more effective than fisetin (130% of control) at preventing the RSL3-induced increase in MCU protein levels.

Overall, the combined data from the calcium assays and the MCU measurement indicate that sterubin and fisetin can prevent RSL3-induced Ca^2+^ overload in mitochondria.

### 3.3. Sterubin and Fisetin Maintain Mitochondrial Biogenesis during Oxytosis/Ferroptosis

The pathophysiology of neurodegenerative diseases is associated with a loss in the mitochondrial biomass [35,36]. The stimulation of mitochondrial biogenesis signaling pathways has revealed beneficial effects in neurodegenerative diseases through the regulation of ROS metabolism and promotion of mitochondrial bioenergetics in order to maintain metabolic homeostasis [35,37]. Thus, we asked whether sterubin’s and fisetin’s homeostatic effects on mitochondrial oxidative stress and Ca^2+^ influx could also involve the preservation or stimulation of mitochondrial biogenesis.

To determine if fisetin and sterubin can affect the mitochondrial content of HT22 cells either alone or following RSL3 treatment, we used the cell-permeant mitochondrion-selective MitoTracker Orange fluorescent probe. A quantitative analysis of HT22 cells (Figure 4A) showed a lower fluorescence intensity of cells treated with RSL3 for 16 h relative to that of control conditions, thereby suggesting a decrease in mitochondrial mass associated with the induction of oxytosis/ferroptosis as previously seen [17]. The co-treatment of RSL3 with sterubin or fisetin preserved mitochondrial mass at the control level. Interestingly, fisetin appeared more potent at counteracting the RSL3-induced mitochondrial mass loss than sterubin. Treatments with sterubin or fisetin alone did not significantly increase mitochondrial mass as compared to control HT22 cells.

In order to investigate mitochondrial biogenesis further, we looked at the effects of the compounds on the AMPK/SIRT1/PGC-1α pathway. Mitochondrial biogenesis is tightly regulated by AMP-activated protein kinase (AMPK), sirtuin-1 (SIRT1), and peroxisome proliferator-activated receptor-γ coactivator 1α (PGC-1α) [38]. Hence, we measured the protein levels of SIRT1, PGC-1α, and the ratio between the phosphorylated AMPK (Thr172) and total AMPK. Western blotting (Figure 4B,C) showed that a 16 h exposure to RSL3 downregulated the AMPK/SIRT1/PGC-1α pathway (*p* < 0.0001). Sterubin alone did not significantly stimulate the AMPK/SIRT1/PGC-1a pathway, consistent with its lack of effect on mitochondrial mass. Fisetin alone did not induce changes in the pAMPK/AMPK ratio or PGC-1α protein levels either. However, the protein expression of SIRT1 was decreased by 33% with fisetin alone as compared to control cells. When HT22 cells were exposed to RSL3 in the presence of sterubin or fisetin, the pAMPK/AMPK ratio and PGC-1α levels were restored to pre-stress conditions (Figure 4C). Curiously, in contrast with fisetin’s negative effect on SIRT1 levels under control conditions, a co-treatment with RSL3 rescued SIRT1 expression relative to RSL3 alone, while a co-treatment with RSL3 and sterubin had a slight but non-significant effect on SIRT1 expression levels compared to RSL3 alone (Figure 4C).

These results suggest that both sterubin and fisetin can maintain mitochondrial biogenesis through the modulation of the AMPK/SIRT1/PGC-1α pathway under conditions that decrease mitochondrial mass such as during oxytosis/ferroptosis, but do not alter mitochondrial biogenesis under control conditions.

### 3.4. Sterubin and Fisetin Regulate Mitochondrial Dynamics and Preserve Mitochondria Networks under Oxytosis/Ferroptosis Conditions

Together with their role in bioenergetics, mitochondria form a dynamic network with the ability to continuously divide, via a mechanism called fission, and collide and fuse with other mitochondria, resulting in fusion. Their morphology is, therefore, highly variable and transient. Alterations in mitochondrial fusion and fission dynamics have been associated with cellular responses to stress (metabolic demand) and caloric intake (metabolic resources) [39].

To study the mitochondrial morphology and dynamics in response to the different treatment conditions, we used a previously generated HT22 cell line with GFP-labelled mitochondria [17]. Following 16 h of treatment with the compounds, the cells were fixed and imaged using Airyscan super-resolution confocal microscopy. As visualized in Figure 5A,B, the control HT22 cells presented elongated and branched mitochondrial networks as previously reported [17,40]. In contrast, the cells treated with RSL3 for 16 h showed more fragmented and shortened mitochondria (Figure 5A,B), similar to what has been seen previously in stressed, pathologic, or aging neurons [17,41,42]. The co-incubation of sterubin or fisetin with RSL3 for 16 h reduced the number of fragmented mitochondria and maintained the elongated mitochondrial network observed in control cells (Figure 5A,B). To further examine the effects of sterubin and fisetin alone and with RSL3 on mitochondrial dynamics, we performed quantitative image analyses of the mitochondrial morphology and networks. Following RSL3 exposure, HT22 cells showed a significant reduction in their mitochondrial summed branch lengths and network branches (Figure 5C,D), indicating a loss of mitochondrial networks concomitant with a shift to small and fragmented structures associated with pathological mitochondrial fission as we reported previously [17]. The exposure of HT22 cells to sterubin or fisetin alone did not significantly alter the length of mitochondrial branches, nor did they affect the count of network branches. However, the co-treatments of sterubin or fisetin with RSL3 did significantly preserve the summed branch lengths and network branches.

Given these results, we, next, measured by Western blotting the protein levels of the mitochondrial fusion proteins optic atrophy protein 1 (OPA1) and mitofusin 2 (MFN2), as well as the mitochondrial fission proteins dynamin-related protein 1 (DRP1) and mitochondrial fission factor (MFF) in the HT22 cells following the different treatments (Figure 5E,F). OPA1, but not MFN2, was upregulated relative to the mitochondrial marker TOM20 following 16 h of treatment with either sterubin or fisetin as compared to control HT22 cells (Figure 5E,G). In contrast, RSL3 dramatically downregulated both OPA1 and MFN2 and these changes in the two fusion proteins were prevented by both sterubin and fisetin. Moreover, the co-incubation of fisetin with RSL3 enhanced the expression of MFN2 as compared to fisetin alone under control conditions. RSL3 also significantly decreased the expression levels of DRP1 and MFF relative to TOM20 (Figure 5F,G). The co-treatments of sterubin or fisetin with RSL3 restored DRP1 to control levels. Interestingly, the co-treatment with sterubin but not fisetin also restored MFF levels following RSL3 treatment.

Overall, the super-resolution microscopy and immunoblotting data suggest that both sterubin and fisetin are effective at protecting against oxytotic/ferroptotic stress-induced alterations in mitochondrial dynamics but, under control conditions, do not appear to have much effect on these dynamics.

### 3.5. Sterubin and Fisetin Enhance Mitochondrial Bioenergetics under Oxytotic/Ferroptotic Stress Conditions

Impairments in mitochondrial bioenergetics and respiration have been shown to precede histological signs of neurodegenerative diseases [8]. Bioenergetics and metabolic regulation are key functions of mitochondria, and mitochondrial respiration via the electron transport chain (ETC) produces ATP to fuel a variety of vital cellular functions. The disruption of mitochondrial bioenergetics can, thus, promote cell damage and death [43].

We used the Seahorse extracellular flux (XF) technology to study the effects of RSL3, sterubin, and fisetin on mitochondrial bioenergetics in HT22 cells. The Seahorse mitochondrial stress test consists of sequentially treating cells with mitochondrial oxidative phosphorylation (OXPHOS) inhibitors (i.e., oligomycin, FCCP, rotenone, and antimycin A). Here, a sublethal concentration of RSL3 (50 nM) was used in order to be able to record data on the responses of mitochondrial respiration to oxytotic/ferroptotic stress in the absence of overt cell death. The results in Figure 6A,B show that RSL3 almost completely inhibited mitochondrial respiration as the oxygen consumption rate (OCR) only slightly responded to the OXPHOS inhibitors, consistent with our previous results [17]. Thus, all of the parameters studied were significantly impaired in the presence of RSL3 alone (Figure 6C–G, *p* < 0.0001). Conversely, the OCR of HT22 cells treated with fisetin or sterubin alone followed the same trend of responses to the different OXPHOS inhibitors as the control cells (Figure 6A,B). However, both fisetin and sterubin decreased basal, as well as maximal, mitochondrial respiration and ATP-linked respiration as compared to control cells. HT22 cells co-treated with RSL3 and sterubin or fisetin for 16 h showed a maintenance of both basal and maximal mitochondrial respiration (Figure 6C,F), and, thereby, ATP-linked respiration as well, as illustrated in Figure 6D. Variations in the proton leak (Figure 6E) revealed that the compounds alone significantly decreased the proton leak compared to control conditions (*p* < 0.0001) and RSL3 exposure was associated with an even further decrease in the proton leak. The co-treatment of RSL3 with sterubin but not fisetin significantly inhibited the proton leak reduction observed with RSL3 alone. Interestingly, the spare respiratory capacity values (Figure 6G) showed that sterubin significantly reduced spare respiratory capacity levels compared to both untreated and fisetin-treated cells (*p* < 0.0001). RSL3 treatment alone yielded a negative spare respiratory capacity. However, the co-treatment of RSL3 with either fisetin or sterubin maintained the spare respiratory capacity at the respective levels of the pre-stress conditions seen with the compounds alone.

Given these results, we, next, examined the ability of the cells to maintain ATP under the various treatment conditions using a chemiluminescent intracellular ATP assay [21]. Our results, shown in Figure 7A, demonstrate a significant drop in ATP levels measured in HT22 cells treated with RSL3 overnight relative to that of control conditions. In contrast, HT22 cells treated with RSL3 and sterubin or fisetin sustained their ATP levels at the control level. Treatments with sterubin or fisetin alone did not significantly increase ATP levels as compared to control HT22 cells. In order to explore the effects of the different treatment conditions on the total ATP production rate and further detect changes in metabolic phenotypes and bioenergetic dependency for ATP production [44] under oxytosis/ferroptosis, we performed the Seahorse Real-Time ATP rate assay, in which cells were sequentially treated with oligomycin and rotenone/antimycin A. Consistent with the total ATP measurements, we observed that sterubin and fisetin alone did not significantly modify the total ATP production rate (Figure 7B). In contrast, RSL3 significantly reduced the ATP production rate, while both sterubin and fisetin rescued the production rate to control levels. Moreover, when we examined the effects of the treatments on ATP production from its two sources, mitochondria (mitoATP) and glycolysis (glycoATP), we found that neither sterubin nor fisetin alone altered the proportion of the ATP production rate from either the mitochondrial or glycolytic pathways under the assay-induced conditions (Figure 7C,D). In contrast, RSL3 exposure significantly reduced the ATP production rate from both mitochondria and glycolysis as compared to control conditions in HT22 cells (*p* < 0.0001, Figure 7C,D). Importantly, both fisetin and sterubin preserved control levels of both mitoATP and glycoATP production rates in the presence of RSL3.

To complete our study on metabolic phenotypes and the various sources of cellular energy in the context of oxytosis/ferroptosis and compound protection, we conducted cellular glycolysis profiling using the Seahorse Glycolysis Stress Test which measures the capacity of the glycolytic pathway after glucose starvation. For these experiments, the HT22 cells were first incubated in the glycolysis stress test medium without glucose or pyruvate to measure the extracellular acidification rate or ECAR. Then, the cells were sequentially treated with saturating concentrations of glucose, oligomycin, and, lastly, 2-deoxy-glucose (2-DG), a glucose analogue that inhibits glycolysis through competitive binding to hexokinase, the first enzyme in the glycolytic pathway. As shown in Figure 8A–C, treatments with fisetin or sterubin alone did not significantly change the ECAR at any point as compared to control conditions. The ECAR was, however, significantly reduced in the HT22 cells following RSL3 exposure (Figure 8A–C). The co-treatment of RSL3 with fisetin maintained the ECAR at control levels but this effect was not significant with sterubin. The glycolytic reserve, shown in Figure 8D, indicates the capacity of a cell to respond to an energetic demand. We did not observe any significant variations in the glycolytic reserve of HT22 cells with fisetin or sterubin treatments alone as compared to control cells. While RSL3 induced a significant loss of the glycolytic reserve as compared to control conditions (*p* < 0.001), sterubin and fisetin increased reserve levels following concomitant RSL3 exposure to those similar to control conditions.

Together, these data demonstrate that both sterubin and fisetin can preserve energy/metabolic homeostasis under RSL3-induced oxytotic/ferroptotic stress conditions in HT22 cells.

### 3.6. Sterubin and Fisetin Protect against Oxytosis/Ferroptosis Independently of Mitochondria

Given the potent anti-oxytotic/ferroptotic properties of sterubin and fisetin, their multiple effects on mitochondria, as well as other targets such as NRF2, and the important role that mitochondrial dysfunction plays in the oxytosis/ferroptosis pathway, we finally asked whether mitochondria were required for the protective effects of the two neuroprotective compounds against RSL3-induced oxytosis/ferroptosis. To accomplish this, we used the previously generated HT22 mt-GFP/mCherry-Parkin cell line overexpressing the ubiquitin E3 ligase Parkin protein, in addition to GFP-labeled mitochondria [17]. As shown previously [45], the overexpression of Parkin, coupled with treatment with mitochondrial uncouplers (e.g., FCCP), activates extensive mitophagy and, thus, can be used to experimentally eliminate mitochondria partially or completely. Figure 9A shows representative fluorescence microscopic images of mitochondria from HT22 mtGFP and mt-GFP/mCherry-Parkin cells treated in parallel in the presence or absence of FCCP. When both cell lines were pre-treated with FCCP, the mt-GFP/mCherry-Parkin cells presented a preserved cellular morphology but with a significant depletion of mitochondria (Figure 9A). In contrast, FCCP did not induce mitochondrial loss in the mt-GFP cells as the GFP-labelled mitochondria were extensively distributed in the soma and their mitochondrial markers were well-retained (Figure 9A). Using the HT22 mt-GFP-expressing cells as a control, we asked whether the protective effects of fisetin and sterubin against oxytosis/ferroptosis persisted after mitochondrial clearance in mt-GFP/mCherry-Parkin-expressing cells. To determine whether our compounds offered comparable protection in the two cell lines in the presence of mitochondria, the mt-GFP or mt-GFP/mCherry-Parkin cells were seeded in parallel at the same density and a DNA-based cell proliferation assay was used to quantitatively assess cell viability. As illustrated in Figure 9B,C, both sterubin and fisetin significantly protected against RSL3 in both the mt-GFP and mt-GFP/mCherry-Parkin cells regardless of whether they were pre-treated with FCCP. To further confirm this result, we also included the mitochondria-specific anti-oxytosis/ferroptosis compound cannabinol (CBN) that has been shown to require functional mitochondria for its protection against RSL3 in the previous work from our lab [17]. As reported previously [17], CBN did not protect against RSL3 in the absence of functional mitochondria (Figure 9B).

These results illustrate that, although fisetin and sterubin strongly preserve mitochondrial health during RSL3-induced oxytosis/ferroptosis, likely contributing to their overall benefits in the context of neurotoxicity, they do not require functional mitochondria to promote cell survival in this context.

## 4. Discussion

Drug development for neurodegenerative diseases is evolving towards targeting pathways relevant to both aging and neurodegeneration, and oxytosis/ferroptosis has been hypothesized to be involved in the pathophysiology of age-related neurodegenerative diseases [9,10,46]. Flavonoids have been proven to be neuroprotective compounds with a potential for slowing neurodegenerative diseases demonstrated by clinical evidence [47,48,49,50,51]. These compounds exhibit multi-target activity and numerous properties such as the modulation of the blood–brain barrier permeability, direct and indirect antioxidant activity, and anti-amyloidogenic and anti-inflammatory properties, making them a valuable source of drug candidates amongst natural products [15,52]. In line with this, we focused on two flavonoids, sterubin and fisetin, for which we showed a robust protective effect against RSL3-induced oxytotic/ferroptotic cell death in HT22 mouse hippocampal neuronal cells. Fisetin and sterubin have both been reported to strongly induce the expression of the antioxidant transcription factor NRF2 [9,13,31]. Importantly, here, we demonstrated that the compounds also upregulate HO-1 and NQO1, two key regulators of the cellular antioxidant defense system directly downstream of NRF2, in the presence of RSL3, as well as in its absence. As HO-1 and NQO1, respectively, control cytoprotection and quinone detoxification to provide antioxidant effects [32,53], the induction of these proteins by our compounds in non-stressed conditions, thus, might provide neurons with higher baseline antioxidant defense levels.

Mitochondrial oxidative stress is a major pathological hallmark of age-related neurodegenerative disorders, as well as a defining feature of the oxytosis/ferroptosis pathway [4,54,55,56]. However, targeting mitochondrial oxidative stress as a strategy against the pathogenesis of neurodegenerative diseases such as AD remains under discussion [36,57]. Our work here shows that both sterubin and fisetin restore mitochondrial redox homeostasis by reducing the amount of mitochondrial ROS induced by RSL3, thus confirming our hypothesis that the compounds prevent ROS production during oxytosis/ferroptosis and, thereby, contribute to the prevention of further intracellular damage. Notably, the small but significant increase in general ROS levels following a 16 h treatment with fisetin or sterubin alone may be associated with their cytoprotective effects such as has previously been observed with some anti-oxytotic/ferroptotic compounds that also decrease lipid peroxidation [58].

Another common hallmark of aging, neurodegenerative diseases, and oxytosis/ferroptosis-induced cell death associated with mitochondrial oxidative stress is increased cellular and mitochondrial Ca^2+^ influx [33,59,60]. This increase is thought to help drive ROS generation, as well as mitochondrial bioenergetic disturbance [61,62]. Fisetin and sterubin modulated mitochondrial Ca^2+^ increases in response to RSL3. We previously found that RSL3-induced oxytosis/ferroptosis led to the upregulation of the mitochondrial calcium uniporter (MCU) [17], a protein described to be involved in the early pathophysiology of neurodegenerative diseases due to its role in the Ca^2+^-dependent membrane potential and bioenergetics [34,62]. Similar to what we reported with CBN [17], sterubin and fisetin both counteracted the stimulating effect of RSL3 on MCU expression, which likely contributes to their ability to maintain mitochondrial Ca^2+^ homeostasis as part of their protective effects against oxytotic/ferroptotic stress.

Although sterubin and fisetin do not stimulate mitochondrial biogenesis under non-stress conditions, we demonstrated for the first time in the HT22 cells that both natural compounds prevent mitochondrial mass loss in RSL3-induced oxytosis/ferroptosis. RSL3 has been shown previously to induce the dysregulation of mitochondrial biogenesis, possibly through interfering with the AMPK/SIRT1/PGC-1α pathway [17], a mechanism shared with the early preclinical stages of age-related neuropathologies [35,37]. Our data show that mitochondrial mass conservation in HT22 nerve cells under oxytotic/ferroptotic stress was associated with the maintenance of the levels of key mitochondrial biogenesis-related proteins by fisetin and sterubin. Interestingly, similar conclusions were reached with regard to the effects of caloric restriction, a robust, pro-longevity intervention, whereby the preservation of mitochondrial function occurred in the absence of mitochondrial biogenesis in aging mouse muscle fibers [63].

At the network level, mitochondrial health and efficiency are modulated by mitochondrial fusion/fission dynamics, a mechanism that, when imbalanced, has been shown to be tightly related to metabolic dysfunction in aging and neurodegenerative diseases [15,64]. Here, again, we found that RSL3 exposure caused mitochondrial fragmentation similar to the one encountered in aging and stress [41,64]. Both compounds maintained network branch counts and summed branch lengths in the presence of RSL3, similar to CBN [17]. Additionally, RSL3 induced the downregulation of the fusion proteins OPA1 and MFN2, and also the fission proteins DRP1 and MFF. The impairment in the dynamic behavior of mitochondria has been found to lead to increased ROS production and reduced ATP production [39]. Both fisetin and sterubin restored the expression levels of inner and outer mitochondrial membrane fusion/fission proteins (OPA1, MFN2, and MFF) as well as DRP1, the key fission protein bridging the outer mitochondrial membrane and the endoplasmic reticulum, thereby preventing fusion/fission dysfunction under oxytosis/ferroptosis induced by RSL3. Sterubin and fisetin may. Thus. preserve the dynamically interconnected tubular and elongated mitochondrial networks that have been associated with bioenergetic efficiency in starvation and longevity [63,65,66]. Furthermore, we found that sterubin and fisetin were potent stimulators of the fusion activator OPA1 under homeostatic conditions. Besides its important role in mitochondrial network shaping with unopposed fusion previously associated with elongated mitochondria [67] and neuroprotective effects [43], OPA1 has also been linked with the preservation of metabolic functions and, more specifically, lower basal respiration rates with stable ATP levels in neurons [68]. In this context, our results imply that sterubin and fisetin may promote mitochondrial phenotypes associated with a higher bioenergetic efficiency.

Impaired mitochondrial bioenergetics have been reported in aging [32], as well as during oxytosis/ferroptosis associated with the AD-associated amyloid β peptide [22]. Interestingly, we found in our bioenergetics study that sterubin and fisetin not only prevented the almost complete dampening of bioenergetics induced by RSL3, but they also lowered basal mitochondrial respiration levels on their own, indicated by lower basal OCRs. This decrease was further accompanied by a concomitant reduction in the OCR linked to ATP production without any significant impairment of overall ATP production or any metabolic phenotype shift in HT22 cells. Collectively, these results suggest that sterubin and fisetin may promote mitochondrial bioenergetic efficiency, in contrast to the effects of RSL3 alone, whereby the dampened OCR and proton leak were associated with reduced ATP production and, ultimately, cell death. Such metabolic features have recently been found in both in vitro and in vivo studies of long-lived organisms [69,70], with increased lifespan metabolism characterized by decreased metabolic rates with sustained ATP levels [70,71]. Moreover, overnight treatments with fisetin and sterubin appear to parallel the metabolic changes induced by a short-term caloric restriction such as a lower proton leak and lower cellular oxygen consumption with maintained ATP production [72,73]. Lowering the proton leak has recently been reported to improve the respiratory function of mitochondria in aged cardiomyocytes [74]. A hypothesis to explain the preservation of mitochondrial function associated with improved energy utilization is the attenuation of oxidative damage by the reduction of ROS production through proton leak modulation [73] coupled with a stimulation of endogenous antioxidant activity. These effects were found following caloric restriction [63] and, as we show here, with the overnight treatment of cells with fisetin or sterubin as well. We speculate, therefore, that the homeostatic effects of sterubin and fisetin on mitochondrial ROS may participate in dampening the decrease in ATP production associated with a proton leak secondary to oxytotic/ferroptotic stress, and this may, ultimately, improve overall energy utilization and mitochondrial physiology. Beyond these effects, the observation that fisetin and sterubin treatments could protect the integrity of rich mitochondrial networks under oxytotic/ferroptotic stress indicates another potential mechanism whereby these compounds optimize the mitochondrial bioenergetic capacity, and is consistent with previous reports [66].

One significant difference between sterubin and fisetin with regard to mitochondrial bioenergetics is illustrated by their effects on maximal OCR and the associated spare respiratory capacity. While fisetin maintained the responsiveness of mitochondrial respiration in HT22 cells at a lower baseline, sterubin seemed to reduce the breadth of the OCR response to acute metabolic demands mimicked by FCCP-induced ETC chain uncoupling. Lower maximal respiration rates similar to those induced by the sterubin overnight treatment were previously described in liver mitochondria and were associated with an increased efficiency in ATP production [75]. The spare respiratory capacity is regarded as an adaptative capacity of mitochondria to respond to cellular metabolic demands [8]. Although a collapse in the spare respiratory capacity has been observed in neurodegenerative diseases, it resulted from maladaptive surges in maximal respiration in response to increased energy requirements induced by cellular stress without the adjustment to basal respiratory levels ultimately leading to bioenergetic exhaustion [8], and this is not consistent with our results here. The spare respiratory capacity should, therefore, be interpreted with caution since values do vary greatly according to cell types and its physiological meaning critically depends on the cellular context. Nonetheless, our data provide clues towards teasing out potential functional variations unique to each of our two neuroprotective compounds. Altogether, these results indicate that the flavonoids sterubin and fisetin not only preserve the integrity of mitochondrial networks, but also promote mitochondrial dynamics to maintain ATP production under oxytotic/ferroptotic stress, and thus improve bioenergetic efficiency.

Strikingly, the two compounds prevented RSL3-induced neuronal death despite the absence of mitochondria. Indeed, both mt-GFP/mCherry-Parkin and mtGFP HT22 cells, treated in parallel, showed the same pattern of cell death when exposed to RSL3 alone and the same pattern of survival when treated with sterubin or fisetin with RSL3 (Figure 9). The finding that the GPx4 inhibitor RSL3 induces cell death in HT22 neurons irrespective of mitochondrial presence is interesting as it provides insights into the contribution of mitochondrial dysfunction to oxytosis/ferroptosis-induced cell death, relative to other factors in the pathway such as lipid peroxidation and calcium influx. Indeed, it implies that oxytosis/ferroptosis cell death induced by the inhibition of GPx4 can occur independently of mitochondrial dysfunction, possibly from an upstream target. Our previous findings from Liang et al. (2022) showed that CBN, a potent neuroprotective and anti-oxytosis-ferroptosis compound, does require functioning mitochondria to protect cells from GPx4-dependent oxytosis/ferroptosis induced by RSL3, and this was replicated again here (Figure 9B,C). Therefore, these results suggest, firstly, that targeting mitochondria regardless of the direct/indirect impact of oxytosis/ferroptosis insults on mitochondria is a promising anti-oxytotic/ferroptotic stress strategy, and, secondly, that sterubin and fisetin differ from the previously described CBN in their protective mechanisms, and, thus, more work should be carried out in the future to further investigate those characteristics.

We found here that the activation of the NRF2 antioxidant axis was the main effect of sterubin and fisetin on HT22 neurons under basal conditions and this is supported by the previous work [7,31]. The recent research places NRF2 as a central modulator of multiple aspects of mitochondrial function such as oxidative stress management, bioenergetics, and biogenesis, particularly with the downstream activation of HO-1 and PGC1α [76]. Therefore, the activation of the NRF2 antioxidant axis could well represent a target through which the compounds sterubin and fisetin protect mitochondria and sustain four main mitochondrial functions (calcium homeostasis, biogenesis, fusion/fission dynamics, and bioenergetics) under oxytotic/ferroptotic stress. More work is required in that regard to identify the respective pathways for sterubin’s and fisetin’s neuroprotective mechanisms. Further, the observation that fisetin and sterubin can provide neuroprotection regardless of mitochondrial presence becomes highly relevant to therapeutic compound identification and drug development as it reveals different possible categories of neuroprotective anti-oxytotic/ferroptotic agents: those, such as CBN, that are specific to mitochondria [17], and those, such as sterubin and fisetin, that have an upstream effect and, thus, modulate not only mitochondria but other cellular targets as well to prevent toxicity and death. As such, compounds specific to mitochondria may be advantageous in early or preventative therapeutic measures, as preventing mitochondrial dysfunction was described as promising with regard to delaying the onset and slowing the progression of neurodegenerative disease and cognitive deficits [43]. Alternatively, in late disease states or pathological forms with high mitochondrial dysfunction, compounds like sterubin or fisetin might be the therapeutic strategy of choice in order to protect the remaining brain tissue alongside the maintenance of mitochondrial integrity and the preservation of cognitive function. Thus, compounds such as fisetin and sterubin that act on mitochondria but also have additional activities independent of mitochondria might hold significant potential as a dual therapy, the likes of which are often seen following the failure of first-line treatments in chronic diseases.

## 5. Conclusions

In conclusion, our research here further confirms the role of the activation of endogenous antioxidant defenses in the protection against oxytotic/ferroptotic stress and highlights markers of mitochondrial dysfunction specific to oxytosis/ferroptosis such as MCU upregulation. The results also provide additional insights into the effects of natural products on mitochondrial pharmacology and how the modulation of mitochondrial resilience, through mitochondrial dynamics and the network architecture, can be associated with bioenergetics to allow protection against cellular stresses, such as the oxytosis/ferroptosis pathway, and improve bioenergetic efficiency as well. Sterubin and fisetin, two pharmacologically distinct flavonoids, reveal, here, different mitochondrially targeted protective mechanisms of action distinct from the previously described neuroprotective agent CBN and, thereby, broaden the therapeutic opportunities for inhibiting oxytosis/ferroptosis. The two flavonoids, thus, continue to hold promise as neuroprotective compounds and therapeutic strategies for age-related neurodegenerative diseases.

## Figures and Tables

**Figure 1 antioxidants-13-00460-f001:**
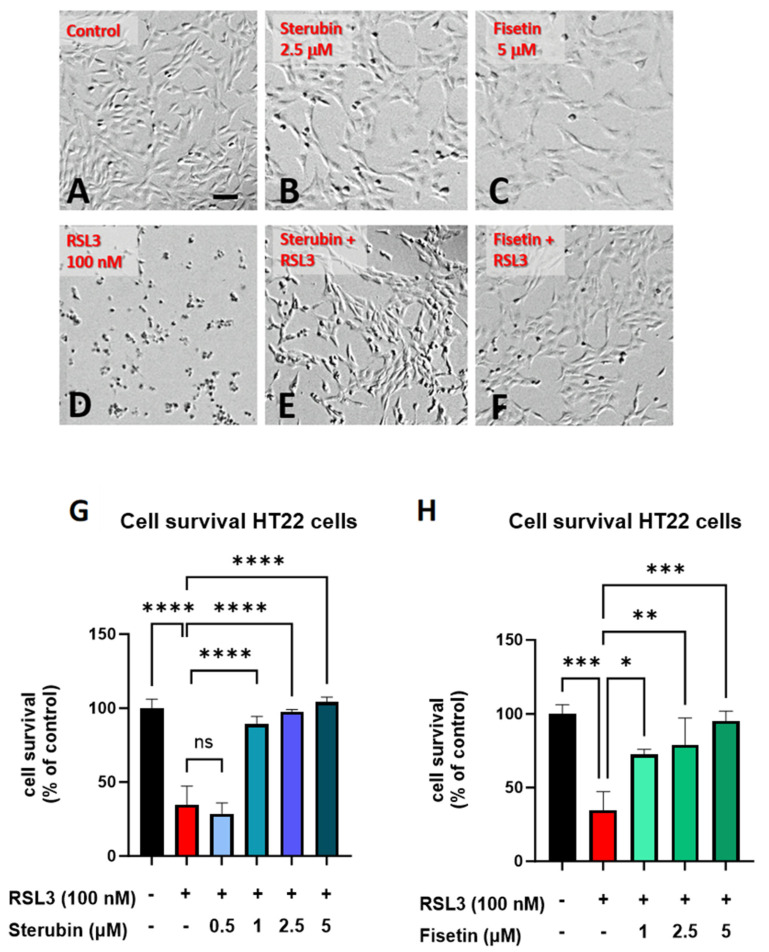
Sterubin and fisetin inhibit oxytosis/ferroptosis in HT22 cells. Representative micrographs of HT22 cells following treatment for 16 h: (**A**) 0.2% ethanol as vehicle control, (**B**) 2.5 μM sterubin, (**C**) 5 μM fisetin, (**D**) 100 nM RSL3, (**E**) 2.5 μM sterubin and 100 nM RSL3, and (**F**) 5 μM fisetin and 100 nM RSL3. Micrographs show the representative morphological characteristics of the cell cultures under a given condition of 16 experimental replicates. Scale bar = 100 μm. (**G**) Cell viability assessment. Cells were treated with 0.2% ethanol vehicle, 100 nM RSL3, and increasing concentrations of sterubin with 100 nM RSL3 for 16 h. (**H**) Cell viability assessment. Cells were treated either with 0.2% ethanol vehicle, 100 nM RSL3, and increasing concentrations of fisetin with 100 nM RSL3 for 16 h. The results are presented as the percentage of the neuroprotective activity relative to control (100%). Data are the mean of 3 replicates per condition ± SD. All data were analyzed by one-way ANOVA with Tukey’s multiple comparison test. * *p* < 0.05, ** *p* < 0.01, *** *p* < 0.001, and **** *p* < 0.0001 relative to control and relative to the 100 nM RSL3 treatment; ns, not significant.

**Figure 2 antioxidants-13-00460-f002:**
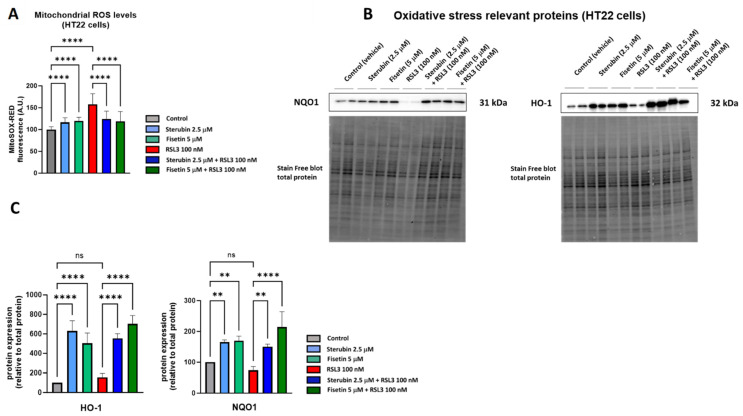
Sterubin and fisetin suppress oxidative stress during oxytosis/ferroptosis in HT22 cells. (**A**) Mitochondrial ROS levels upon different treatment conditions in the cells for 16 h. Data were normalized to total protein/well and are the percentages of fluorescence relative to control of the mean of 16 replicates per condition from 4 independent experiments ± SD. (**B**) Western blot data of HO-1 and NQO1 and total protein (*n =* 3–4). Protein levels were measured upon different treatment conditions of the cells for 16 h. (**C**) Densitometric quantification of the Western blots. Data were normalized to total protein and are percentages relative to control of the mean ± SD. All data were analyzed by one-way ANOVA with Tukey’s multiple comparison test. ** *p* < 0.01, and **** *p* < 0.0001 relative to control and relative to the 100 nM RSL3 treatment; ns, not significant.

**Figure 3 antioxidants-13-00460-f003:**
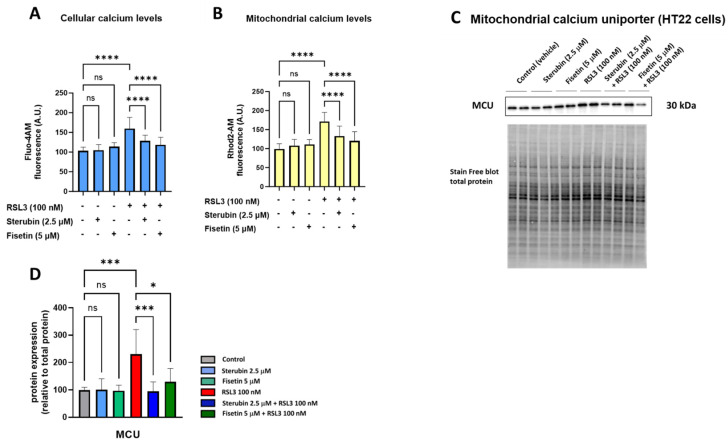
Fisetin and sterubin maintain Ca^2+^ homeostasis during oxytosis/ferroptosis in HT22 cells. (**A**) Cellular Ca^2+^ levels upon different treatment conditions of the cells for 16 h. Data were normalized to total protein/well and are the mean of 4–12 replicates per condition from 4 independent experiments ± SD. (**B**) Mitochondrial Ca^2+^ levels upon different treatment conditions of the cells for 16 h. Data were normalized to total protein/well and are the mean of 4–12 replicates per condition from 4 independent experiments ± SD. (**C**) Western blot data of MCU and total protein (*n =* 3). Protein levels were measured following different treatments of the cells for 16 h. (**D**) Densitometric quantification of the Western blots. Data were normalized to total protein and are percentages relative to control of the mean ± SD. All data were analyzed by one-way ANOVA with Tukey’s multiple comparison test. * *p* < 0.05, *** *p* < 0.001, and **** *p* < 0.0001 relative to control and relative to the 100 nM RSL3 treatment; ns, not significant.

**Figure 4 antioxidants-13-00460-f004:**
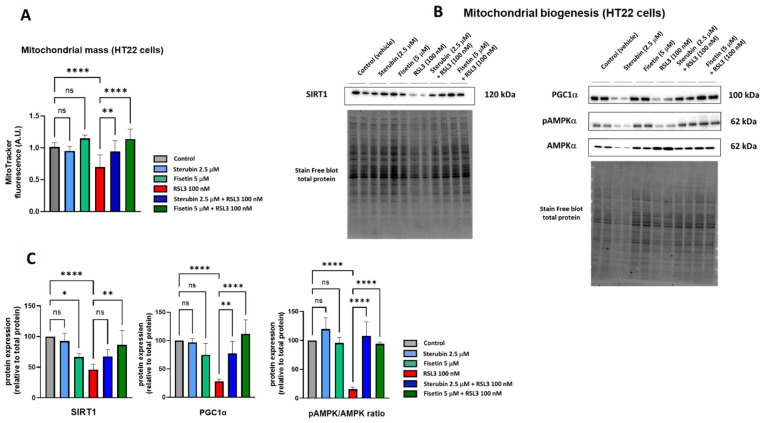
Sterubin and fisetin maintain mitochondrial biogenesis following induction of oxytosis/ferroptosis in HT22 cells. (**A**) Relative quantification of mitochondrial mass with MitoTracker fluorescence in HT22 cells following different treatment conditions for 16 h. Data were normalized to total protein/well and are the means of 10 replicates per condition relative to control ± SD. (**B**) Western blot data of SIRT1, PGC-1α, pAMPKα (Thr172), and total AMPKα and respective total protein (*n =* 4). Protein levels were measured following the different treatments of the cells for 16 h. (**C**) Densitometric quantification of the Western blots. Data were normalized to total protein and are percentages relative to control of the mean ± SD. All data were analyzed by one-way ANOVA with Tukey’s multiple comparison test. * *p* < 0.05, ** *p* < 0.01, and **** *p* < 0.0001 relative to control and relative to the 100 nM RSL3 treatment; ns, not significant.

**Figure 5 antioxidants-13-00460-f005:**
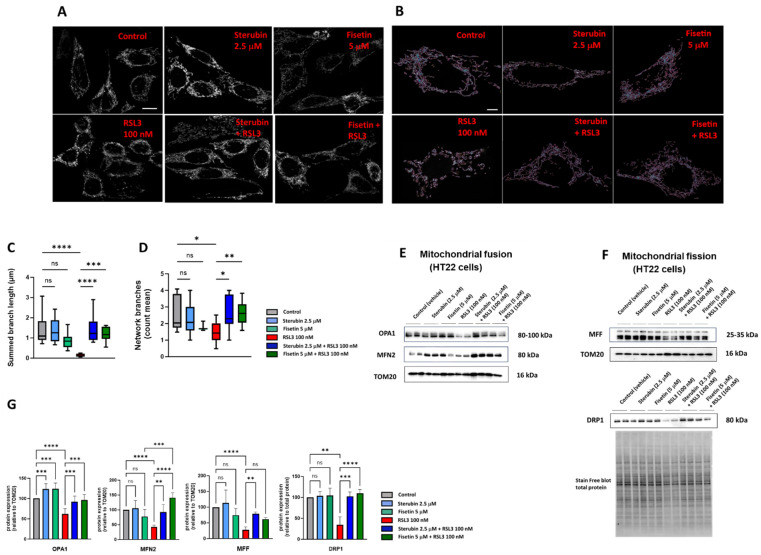
Sterubin and fisetin regulate mitochondrial fusion/fission dynamics following induction of oxytosis/ferroptosis in HT22 cells. (**A**) Representative fluorescent images of HT22 mt−GFP cells following treatment for 16 h: (**top left**) 0.2% ethanol as vehicle control, (**top middle**) 2.5 μM sterubin, (**top right**) 5 μM fisetin, (**bottom left**) 100 nM RSL3, (**bottom middle**) 2.5 μM sterubin and 100 nM RSL3, and (**bottom right**) 5 μM fisetin and 100 nM RSL3. The micrographs show representative images of the morphological characteristics of the cells under the different conditions with 3 experimental replicates per condition. Scale bar = 10 μm. (**B**) Representative images of mitochondrial skeletons from the MINA Plugin of HT22 mt-GFP cells following treatment for 16 h used for quantification: (**top left**) 0.2% ethanol as vehicle control, (**top middle**) 2.5 μM sterubin, (**top right**) 5 μM fisetin, (**bottom left**) 100 nM RSL3, (**bottom middle**) 2.5 μM sterubin and 100 nM RSL3, and (**bottom right**) 5 μM fisetin and 100 nM RSL3. Scale bar = 5 μm. (**C**,**D**) Mitochondrial network morphology analyses on micrographs of HT22 cells after different treatments: (**C**) mitochondrial summed branch length, and (**D**) mitochondrial network branches. Data are the mean of 12 cells per condition ± SD. (**E**,**F**) Western blot data of OPA1, MFN2, DRP1, MFF, TOM20, and stain-free blot total protein (*n =* 3–5). Protein levels were measured following the different treatments of the cells for 16 h. (**G**) Densitometric quantification of the Western blots. Data were normalized to total protein and TOM20 except for DRP1 normalized to total protein and are percentages relative to control of the mean ± SD. All data were analyzed by one-way ANOVA with Tukey’s multiple comparison test. * *p* < 0.05, ** *p* < 0.01, *** *p* < 0.001, and **** *p* < 0.0001 relative to control and relative to the 100 nM RSL3 treatment; ns, not significant.

**Figure 6 antioxidants-13-00460-f006:**
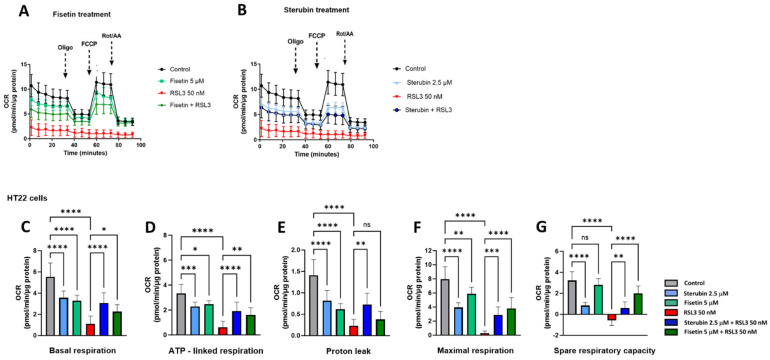
Sterubin and fisetin preserve mitochondrial bioenergetics following induction of oxytosis/ferroptosis in HT22 cells. (**A**,**B**) Mitochondrial oxygen consumption rate (OCR) profiles in HT22 cells after different treatments for 16 h. (**A**) OCR following fisetin treatment. (**B**) OCR following sterubin treatment. (**C**–**G**) Graphs for basal respiration, OCR linked to ATP production, OCR linked to proton leak, and maximal respiration and spare respiratory capacity in HT22 cells. All data were normalized to total protein/well and are the mean of 16 replicates per condition ± SD. All data were analyzed by one-way ANOVA with Tukey’s multiple comparison test. * *p* < 0.05, ** *p* < 0.01, *** *p* < 0.001, and **** *p* < 0.0001 relative to control and relative to the 50 nM RSL3 treatment; ns, not significant.

**Figure 7 antioxidants-13-00460-f007:**
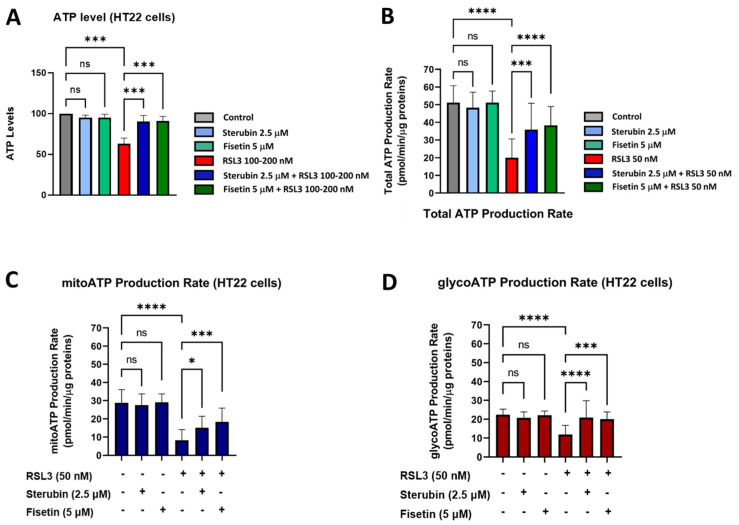
Sterubin and fisetin preserve ATP production and the metabolic phenotype of HT22 cells following induction of oxytosis/ferroptosis. (**A**) ATP levels in HT22 cells treated with 2.5 μM sterubin, 5 μM fisetin, 100–200 nM RSL3, 2.5 μM sterubin and 100–200 nM RSL3, 5 μM fisetin and 100–200 nM RSL3. The results are presented as the percentage of ATP levels relative to control (100%). Data are the mean of 3–4 replicates per condition ± SD. (**B**–**D**) Graphs for total ATP production rate, mitoATP production rate, and glycoATP production rate in HT22 cells. Data were normalized to total protein/well and are the mean of 16 replicates per condition ± SD. All data were analyzed by one-way ANOVA with Tukey’s multiple comparison test. * *p* < 0.05, *** *p* < 0.001, and **** *p* < 0.0001 relative to control and relative to the 50 and 100–200 nM RSL3 treatment; ns, not significant.

**Figure 8 antioxidants-13-00460-f008:**
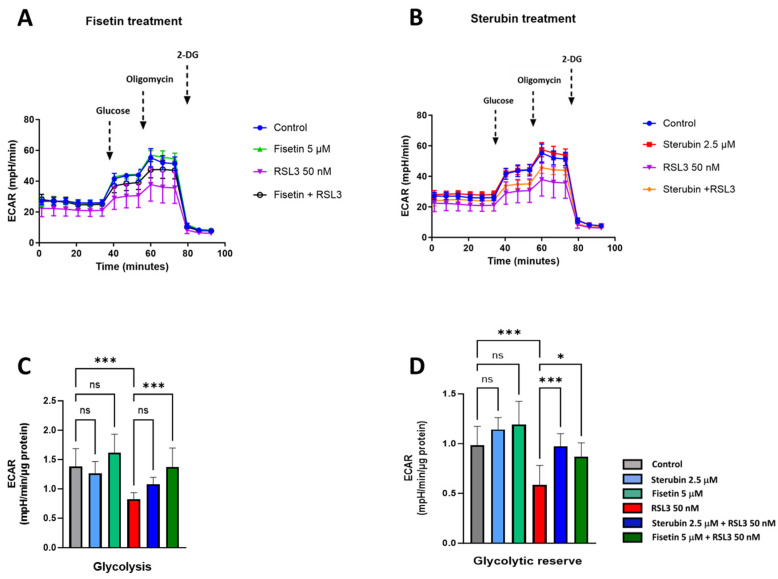
Sterubin and fisetin preserve glycolytic rate following induction of oxytosis/ferroptosis in HT22 cells. (**A**,**B**) Extracellular acidification rate (ECAR) profiles in HT22 cells after different treatments for 16 h. (**A**) ECAR following fisetin treatment. (**B**) ECAR following sterubin treatment. (**C**,**D**) Graphs for glycolytic rate and glycolytic reserve in HT22 cells. All data were normalized to total protein/well and are the mean of 16 replicates per condition ± SD. All data were analyzed by one-way ANOVA with Tukey’s multiple comparison test. * *p* < 0.05, and *** *p* < 0.001 relative to control and relative to the 50 nM RSL3 treatment; ns, not significant.

**Figure 9 antioxidants-13-00460-f009:**
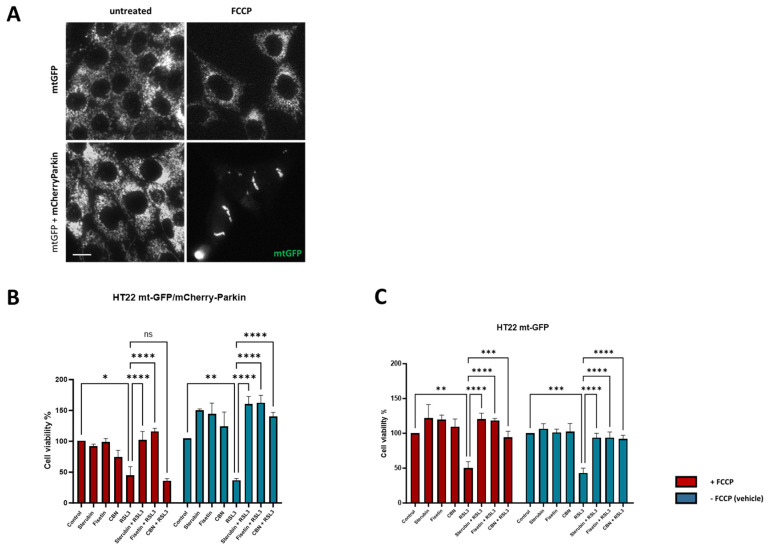
Sterubin and fisetin protect against oxytosis/ferroptosis in HT22 cells independently of mitochondria. (**A**) Representative green fluorescent images of mitochondria from HT22 mtGFP and mt-GFP/mCherry-Parkin cells in the presence or absence of FCCP (10 μM). Scale bar = 10 μm. (**B**,**C**) DNA-based cell viability analysis of HT22 mt–GFP/mCherry-Parkin or mt-GFP cells following the different treatment conditions: 0.2% ethanol as vehicle control, 2.5 μM sterubin, 5 μM fisetin, 5 μM CBN, 100 nM RSL3, 2.5 μM sterubin and 100 nM RSL3, 5 μM fisetin and 100 nM RSL3, 5 μM CBN and 100 nM RSL3. Data are percentages of the mean of 3 replicates per condition ±SD. All data were analyzed by one-way ANOVA with Tukey’s multiple comparison test. * *p* < 0.05, ** *p* < 0.01, *** *p* < 0.001, and **** *p* < 0.0001 relative to control and relative to the 100 nM RSL3 treatment; ns, not significant.

## Data Availability

The datasets generated in the current study are available from the corresponding author upon reasonable request. The data are not publicly available due to [The data are in a multitude of formats which are not appropriate for upload to a public platform however they are available from the corresponding author upon reasonable request].

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
