# Peer review of "The Neuroprotective Flavonoids Sterubin and Fisetin Maintain Mitochondrial Health under Oxytotic/Ferroptotic Stress and Improve Bioenergetic Efficiency in HT22 Neuronal Cells"

_antioxidants, 2024, doi:10.3390/antiox13040460_

Round 1
Reviewer 1 Report
The study investigated the neuroprotective effects of flavonoids sterubin and fisetin against RSL3-induced oxytotic/ferroptotic stress in HT22 cells and was mainly focused on mitochondrial physiology. The authors showed that these two flavonoids preserve mitochondrial functions, and that mitochondria are not required for the observed neuroprotective effects. The research design is appropriate. The obtained results are interesting and may help in the development of novel therapeutic strategies against aging and age-related neurodegenerative disorders. They also provide deeper understanding of mitochondrial dysfunction during oxytosis/ferroptosis. In my opinion, some results just need to be further discussed. Experiments on another cell line or with another oxidative stress-inducer would strengthen the conclusions.
For better clarity, the authors should explain the following issues:
1. How the prooxidative effects of fisetin and sterubin in physiological conditions may be associated with their cytoprotective effects during oxytotic/ferroptotic stress? What about the potential adverse effects of increased ROS in physiological conditions? (L302, L644)
2. L310 – ROS increase in mitochondria in physiological conditions was accompanied with the prominent induction of antioxidative genes HO-1 and NQO1. Please comment as in stress conditions upregulation of HO-1 and NQO1 was accompanied with ROS decrease.
3. L632 - “stimulation by our compounds in non-stressed conditions thus might provide neurons with a lower baseline level of oxidative stress.” – this conclusion seems incorrect as ROS were increased
4. What type of death is oxytosis/ferroptosis? Why RSL3 is used instead of high concentrations of glutamate in neuronal cells? Why (at least some) of the results were not confirmed with glutamate in HT22 cells or with another neuronal cells (eg. differentiated SH-SY5Y) exposed to RSL3?
5. One of the main conclusions of the study is that effects of sterubin and fisetin were independent of mitochondria. The protective effect was demonstrated in the absence of functional mitochondria (“persisted after mitochondrial clearence”). What about the absolute viability of cells in the absence of functional mitochondria (HT22 mtGFP compared to mt-GFP/mCherry-Parkin cells in the presence of FCCP)? How were cells able to respond to RSL3 in the absence/prominent depletion of ATP? There is a great possibility that death pathways of mt-GFP/mCherry-Parkin cells in the presence of FCCP and RSL3 differ from the death pathways initiated in wt HT22 cells exposed only to RSL3 (perhaps due to the presence of necrotic processes). All this makes comparison and conclusions difficult. The conclusion that mitochondria are not required for the neuroprotective effects of sterubin and fisetin should be discussed more convincingly. The photographs of these cells should be provided.
Here are also some minor comments:
L42 – ref. 5 not appropriate
L125 – please add brief explanation for FCCP
L175 – ETC -please explain abbreviation
L188 – ECAR – the same as above
L299 – „showed a 58% increase in the MitoSOX signal in comparison to the control or flavonoid-treated cells (p <0.0001).” – increase is not 58% in comparison to flavonoid-treated cells
Figure 2, legend – *p < 0.05 could be omitted
Figures 3 and 6, legend – **p < 0.01 could be omitted
Figure 4, legend – ***p < 0.001 could be omitted
Figure 5, legend – TOM20 should be mentioned in the main text
Figure 4 – photographs with MitoTracker Orange fluorescent probe should be added
Why different colour patterns were used for 7C and 7D?
Reference list – journal names should be abbreviated
Author Response
Reviewer 1:
- How the prooxidative effects of fisetin and sterubin in physiological conditions may be associated with their cytoprotective effects during oxytotic/ferroptotic stress? What about the potential adverse effects of increased ROS in physiological conditions? (L302, L644)
A: Thank you for this comment. ROS play essential roles in various physiological processes such as epigenetics or cell signalling and, more particularly, NRF2-induced antioxidant signalling pathways via KEAP1 release (Lennicke et al., 2021). It is an excessive accumulation of ROS that can lead to oxidative stress, for example under stress conditions such as RSL3 toxicity. Fisetin and sterubin induce antioxidant cellular defenses through NRF2 activation and subsequent NQO1 and HO-1 upregulation, which help to maintain redox homeostasis. Additionally, our bioenergetics study on HT22 cells shows that both flavonoids reduce baseline respiration rate under physiological conditions and reduce the proton leak which, when elevated, could be associated with a surge in ROS. Overall, the pro-oxidative effects of fisetin and sterubin in physiological conditions observed with MitoSOX-RED fluorescence in HT22 cells (Figure 2) seem to be counteracted by their larger mitochondrial and cellular antioxidant effects.
- L310 – ROS increase in mitochondria in physiological conditions was accompanied with the prominent induction of antioxidative genes HO-1 and NQO1. Please comment as in stress conditions upregulation of HO-1 and NQO1 was accompanied with ROS decrease.
A: Thank you for your comment.
Under physiological conditions, HO-1 and NQO-1 are upregulated by both sterubin or fisetin as demonstrated following flavonoid treatment of HT22 cells in comparison to control cells (Figure 2C). The increase in expression levels of these two antioxidant proteins is believed to be secondary to NRF2 activation as previously demonstrated by our group (Fischer et al., 2019 & Ehren et al., 2013).
Under RSL3 stress, the oxytosis/ferroptosis cell death pathway is activated and an important aspect of this cell death pathway is illustrated in Figure 2A which shows a surge in mitochondrial ROS relative to control conditions. Furthermore, the HT22 cells exposed to RSL3 stress exhibit no significant change in the expression levels of HO-1 or NQO1 as compared to control conditions (Figure 2C).
Under RSL3 stress conditions, sterubin and fisetin treatment are still associated with high levels of antioxidative protein expression as compared to control conditions (Figure 2C). However, Figure 2A shows that the ROS surge caused by RSL3 in HT22 cells is significantly dampened by the flavonoid treatment and this is most likely to be secondary to activation of cellular antioxidant systems as illustrated by the expression levels of HO-1 and NQO1 (Figure 2C).
- L632 - “stimulation by our compounds in non-stressed conditions thus might provide neurons with a lower baseline level of oxidative stress.” – this conclusion seems incorrect as ROS were increased
A: Thank you for your comment. We have made the following correction to this statement so it is not in contradiction with other results discussed: “As HO-1 and NQO1 respectively control cytoprotection and quinone detoxification to provide antioxidant effects [32,53], induction of these proteins by our compounds in non-stressed conditions thus might provide neurons with higher baseline antioxidant defense levels.’’
- What type of death is oxytosis/ferroptosis? Why RSL3 is used instead of high concentrations of glutamate in neuronal cells? Why (at least some) of the results were not confirmed with glutamate in HT22 cells or with another neuronal cells (eg. differentiated SH-SY5Y) exposed to RSL3?
A: Oxytosis/ferroptosis is a type of non-apoptotic regulated cell death pathway dependent on intracellular calcium influx (Tan et al., 2001, Maher et al., 2020).
RSL3 is a downstream inducer of oxytosis/ferroptosis, through inhibition of glutathione peroxidase 4 (GPx4), while glutamate acts on the upstream system xc- to trigger the oxytosis/ferroptosis pathway via glutathione depletion. We have previously reported on the effects of fisetin and sterubin against glutamate toxicity (Fischer et al., 2019 & Ehren et al., 2013) so here we wanted to focus on their effects against RSL3 toxicity. In addition, this study supports and extends our recent work on CBN (Liang et al., 2022) where RSL3 was also used as the primary oxytosis/ferroptosis inducer and we focused on mitochondrial dysfunction.
Some of the experiments were also conducted with erastin, a small molecule found to be an inhibitor of system xc-. However, only the results with RSL3 were included in this manuscript in order to reduce the number of variables and conditions and keep the manuscript more focused.
We exclusively used HT22 cells for this study as we have done in many previous papers. These cells are highly sensitive to oxytosis/ferroptosis and thus provide an excellent model for examining this cell death pathway in cell culture. However, we acknowledge that not including another cell type could be considered a limitation of this study.
- One of the main conclusions of the study is that effects of sterubin and fisetin were independent of mitochondria. The protective effect was demonstrated in the absence of functional mitochondria (“persisted after mitochondrial clearence”). What about the absolute viability of cells in the absence of functional mitochondria (HT22 mtGFP compared to mt-GFP/mCherry-Parkin cells in the presence of FCCP)? How were cells able to respond to RSL3 in the absence/prominent depletion of ATP? There is a great possibility that death pathways of mt-GFP/mCherry-Parkin cells in the presence of FCCP and RSL3 differ from the death pathways initiated in wt HT22 cells exposed only to RSL3 (perhaps due to the presence of necrotic processes). All this makes comparison and conclusions difficult. The conclusion that mitochondria are not required for the neuroprotective effects of sterubin and fisetin should be discussed more convincingly. The photographs of these cells should be provided.
A: Thank you for these comments.
The absolute viability of the different HT22 cell lines in the absence of functional mitochondria was not investigated here as the scope of this study is mitochondrial health under oxytosis/ferroptosis stress conditions. However, both cell types were treated in parallel at the same timepoints which means that both mt-GFP/mCherry-Parkin and mtGFP HT22 cells were alive at 5h after mitochondrial uncoupling with FCCP prior to being exposed to RSL3 to induce oxytosis/ferroptosis. Importantly, we have previously shown (Liang et al, 2022) that a well-characterized inhibitor of oxytosis/ferroptosis, the radical trapping antioxidant ferrostatin, protects the cells from RSL3 toxicity following removal of mitochondria, strongly supporting the idea that the cells still die from oxytosis/ferroptosis under these conditions.
The finding that the GPx4 inhibitor RSL3 induces cell death in HT22 neurons irrespective of mitochondrial presence is interesting as it provides insights into the contribution of mitochondrial dysfunction to oxytosis/ferroptosis-induced cell death, relative to other factors in the pathway such as lipid peroxidation and calcium influx. Indeed, it implies that oxytosis/ferroptosis cell death induced by inhibition of GPx4 can occur independently of mitochondrial dysfunction. Nonetheless, our previous findings from Liang et al. (2022) showed that CBN, a potent neuroprotective and anti-oxytosis-ferroptosis compound, does require functioning mitochondria to protect cells from GPx4-dependent oxytosis/ferroptosis induced by RSL3, and this was replicated again here (Figure 9). Therefore, these results suggest firstly that targeting mitochondria regardless of the direct/indirect impact of oxytosis/ferroptosis insults on mitochondria, is a promising anti-oxytotic/ferroptotic stress strategy and secondly, that sterubin and fisetin differ from the previously described CBN in their protective mechanisms and thus, more work should be done in the future to further investigate those characteristics.
Representative green fluorescent images of mitochondria from HT22 mtGFP and mt-GFP/mCherry-Parkin cells in the presence or absence of FCCP are provided in Figure 9A. An additional figure (below) was made to illustrate quantitative data. Representative fluorescent greyscale images of mitochondria from HT22 mt-GFP/mCherry-Parkin after 5h of FCCP treatment and following 7h of treatment with experimental conditions. (top left) 0.2% ethanol as vehicle control, (top middle) 2.5 μM sterubin, (top right) 5 μM fisetin, (bottom left) 100 nM RSL3, (bottom middle) 2.5 μM sterubin and 100 nM RSL3, (bottom right) 5 μM fisetin and 100 nM RSL3. The micrographs show representative images of the morphological characteristics of the cells under the different conditions with 3 replicates per condition. Scale bar = 10 μm.
''Figure in attached file.''
Here are also some minor comments:
L42 – ref. 5 not appropriate
L125 – please add brief explanation for FCCP
L175 – ETC -please explain abbreviation
L188 – ECAR – the same as above
L299 – „showed a 58% increase in the MitoSOX signal in comparison to the control or flavonoid-treated cells (p <0.0001).” – increase is not 58% in comparison to flavonoid-treated cells
Figure 2, legend – *p < 0.05 could be omitted
Figures 3 and 6, legend – **p < 0.01 could be omitted
Figure 4, legend – ***p < 0.001 could be omitted
Figure 5, legend – TOM20 should be mentioned in the main text
Reference list – journal names should be abbreviated
A: Thank you for these suggestions. We have included those corrections in the revised manuscript and they can be found highlighted in the manuscript.
Figure 4 – photographs with MitoTracker Orange fluorescent probe should be added
A: Thank you for this comment. The quantitative data were tailored for microplate reading, which is not suitable for microscopic imaging.
Why different colour patterns were used for 7C and 7D?
A: Different colour patterns were used in those graphs to better differentiate the different sources of ATP produced.
Sources:
- Lennicke, C., & Cochemé, H. M. (2021). Redox metabolism: ROS as specific molecular regulators of cell signaling and function. Molecular cell, 81(18), 3691–3707.
- Khan N, Syed DN, Ahmad N, Mukhtar H. Fisetin: a dietary antioxidant for health promotion. Antioxid Redox Signal. 2013;19(2):151-162.
- Tan, S.; Schubert, D.; Maher, P. Oxytosis: a novel form of programmed cell death. Curr Top Med Chem. 2001, 1, 497–506.
- Fischer, W.; Currais, A.; Liang, Z.; Pinto, A.; Maher, P. Old age-associated phenotypic screening for Alzheimer's disease drug candidates identifies sterubin as a potent neuroprotective compound from Yerba santa. Redox Biol. 2019, 21, 101089.
- Liang, Z.; Soriano-Castell, D.; Kepchia, D.; Duggan, B. M.; Currais, A.; Schubert, D.; Maher, P. Cannabinol inhibits oxytosis/ferroptosis by directly targeting mitochondria independently of cannabinoid receptors. Free Radic Biol Med. 2022, 180, 33–51.
- Maher, P.; Currais, A.; Schubert, D. Using the Oxytosis/Ferroptosis Pathway to Understand and Treat Age-Associated Neurodegenerative Diseases. Cell Chem Biol. 2020, 27(12), 1456-1471.
- Davis, J. B.; Maher, P. Protein kinase C activation inhibits glutamate-induced cytotoxicity in a neuronal cell line. Brain res. 1994, 652(1), 169–173.
- Li, Y., Maher, P., & Schubert, D. (1997). Requirement for cGMP in nerve cell death caused by glutathione depletion. J cell biol, 139(5), 1317–1324.
- Tan, S., Sagara, Y., Liu, Y., Maher, P., & Schubert, D. (1998). The regulation of reactive oxygen species production during programmed cell death. J cell biol, 141(6), 1423–1432.

Reviewer 2 Report
The authors used two different natural compounds and examined their neuroprotective effects in oxytotic/ferroptotic stress. They carried out many experiments to examine many aspects of mitochondrial behavior under stress conditions and with the co-administration of the two flavonoids. They checked the amount of mitochondria at different treatments, as well as the ROS generation, the Ca-ion movement within the neuron. They looked at the biogenesis of mitochondria, their fission-fusion activities together with expression levels of some proteins involved in the above processes. They watched the ATP production of the cells and some signaling pathways activities. Interestingly, the neuroprotective effects of the two flavonoids were partially independent of the presence of functioning mitochondria. They examined many aspects of the question, and reached new consequences, which are useful in clinical practice.
My questions: Did the authors have experiments when the flavonoid administration followed the stress treatment? If yes, what were the results? What are the concentration of these compounds in their natural sources?
One technical remark: the text in the colored column graphs is almost unreadable, the letters are two small.
Author Response
Reviewer 2:
- Did the authors have experiments when the flavonoid administration followed the stress treatment? If yes, what were the results?
A: Thank you for your suggestion. No, we did not conduct those experiments in this study. However, these types of experiments are worth doing in future studies. Previous work has shown that oxytosis/ferroptosis cell death begins from 6-8 h post treatment with glutamate or cystine deprivation (Davis and Maher, 1994; Li et al., 1997; Tan et al.,1998). However, once the death cascade is fully activated, no anti-oxytosis/ferroptosis compounds that we have tested were able to rescue the cells.
- What are the concentration of these compounds in their natural sources?
A: Thank you for this question. Sterubin is found at low concentrations in the Californian plant Yerba santa (Eriodictyon californicum) although the levels vary tremendously depending on the growth conditions and can be up to 1% of the dry weight of the leaf (Maher et al., Front. Pharmacol. 2020). The concentration of fisetin in strawberries has been published as 160 μg/g (Khan et al., 2013). Note however, that both our compounds used in this study were synthesized for research purposes. “Sterubin was obtained from Michael Decker [19], fisetin was obtained from Indofine” from Materials & methods section.
- One technical remark: the text in the colored column graphs is almost unreadable, the letters are two small.
A: Thank you for your suggestion. The size of the text has now been increased in all figures of the revised manuscript.
Sources:
- Lennicke, C., & Cochemé, H. M. (2021). Redox metabolism: ROS as specific molecular regulators of cell signaling and function. Molecular cell, 81(18), 3691–3707.
- Khan N, Syed DN, Ahmad N, Mukhtar H. Fisetin: a dietary antioxidant for health promotion. Antioxid Redox Signal. 2013;19(2):151-162.
- Tan, S.; Schubert, D.; Maher, P. Oxytosis: a novel form of programmed cell death. Curr Top Med Chem. 2001, 1, 497–506.
- Fischer, W.; Currais, A.; Liang, Z.; Pinto, A.; Maher, P. Old age-associated phenotypic screening for Alzheimer's disease drug candidates identifies sterubin as a potent neuroprotective compound from Yerba santa. Redox Biol. 2019, 21, 101089.
- Liang, Z.; Soriano-Castell, D.; Kepchia, D.; Duggan, B. M.; Currais, A.; Schubert, D.; Maher, P. Cannabinol inhibits oxytosis/ferroptosis by directly targeting mitochondria independently of cannabinoid receptors. Free Radic Biol Med. 2022, 180, 33–51.
- Maher, P.; Currais, A.; Schubert, D. Using the Oxytosis/Ferroptosis Pathway to Understand and Treat Age-Associated Neurodegenerative Diseases. Cell Chem Biol. 2020, 27(12), 1456-1471.
- Davis, J. B.; Maher, P. Protein kinase C activation inhibits glutamate-induced cytotoxicity in a neuronal cell line. Brain res. 1994, 652(1), 169–173.
- Li, Y., Maher, P., & Schubert, D. (1997). Requirement for cGMP in nerve cell death caused by glutathione depletion. J cell biol, 139(5), 1317–1324.
- Tan, S., Sagara, Y., Liu, Y., Maher, P., & Schubert, D. (1998). The regulation of reactive oxygen species production during programmed cell death. J cell biol, 141(6), 1423–1432.
